# OpenReview forum: "More RLHF, More Trust? On The Impact of Preference Alignment On Trustworthiness"
_ICLR.cc/2025/Conference — ICLR 2025 Oral_

### Official Review · Reviewer_p6uw · 2024-10-26

**Soundness:** 3
**Presentation:** 4
**Contribution:** 3
**Rating:** 6
**Confidence:** 3

**Summary:**

The work investigates the impact of RLHF on the trustworthiness of LLMs. evaluating key dimensions such as toxicity, stereotypical bias, machine ethics, truthfulness, and privacy. Findings indicate that RLHF does not consistently enhance trustworthiness and may even degrade it in some areas.

Further analysis with targeted datasets and a discussion of potential solutions would strengthen its impact. However, this contribution is significant and could be accepted for the conference.

**Strengths:**

- **Well-structured:** The paper is organized clearly and uses precise language. It offers the first systematic evaluation of RLHF’s impact on specific aspects of trustworthiness.
- **Thorough Evaluation:** Multiple trustworthiness dimensions of RLHF’s effects are thoroughly examined.
- **Innovative Methodology:** The application of influence-function-based data attribution methods to RLHF provides valuable insights into how training data influences model behavior.
- **Empirical Rigor:** The results are detailed, with comprehensive benchmarks and rigorous experimental designs, involving multiple models and controlled benchmarks, allowing for broader generalization.

**Weaknesses:**

- **Mitigation Strategies Lacking:** Although RLHF exacerbates bias and privacy issues, potential solutions are not explored in depth. There is limited discussion on why these misalignments occur and how they could be systematically addressed.
- **Dataset Limitations:** The reliance on the Anthropic HH dataset for RLHF may limit the generalizability of findings across different applications and LLM architectures. The general-purpose nature of this dataset appears somewhat misaligned with the targeted trustworthiness benchmarks, potentially skewing the results.
- **Model Scale Constraints:** The analysis is limited to models of up to Pythia-6.9B and Llama-7B, which may not reflect performance in larger, frequently used LLMs.
- **Figure Clarity:** Certain figures, such as Figures 5 and 11, are not optimally clear. Text and data appear crowded, reducing readability.
- **Dense Methodology Sections:** Sections such as 3 and 5 contain dense explanations that could benefit from clearer, more accessible language.

**Questions:**

- Can additional fine-tuning methods be combined with RLHF to mitigate observed trustworthiness degradation?
- Would using smaller, targeted RLHF datasets for each trustworthiness aspect outperform a single general-purpose dataset? Additionally, how would the combination of general-purpose and aspect-specific datasets impact trustworthiness?
- How does dataset size influence the observed trends in trustworthiness metrics? Specifically, would varying dataset sizes in RLHF lead to different levels of model trustworthiness?

---

> ### Author Response · Authors · 2024-11-19
> **Author Rebuttal for Reviewer p6uw**
>
> We genuinely appreciate your feedback on our paper, and here we would like to address your questions and concerns raised in the review. We also include our responses for common questions shared by multiple reviewers in a separate section above.
>
> $\textbf{Q4.1 The cause of the misalignment issue and potential mitigation strategies}$
>
> $\textbf{A4.1}$
>
> Thank you for your feedback. We addressed a similar question in A2.2, and we encourage you to refer to it for additional context. The purpose of section 5 in our paper is to explain the misalignment issue from a data-centric perspective, where we propose adapting data attribution methods to the RLHF setting. This approach enables users to identify influential data points that may negatively impact trustworthiness and provides a pathway to address these issues through dataset pruning.
>
> However, we acknowledge that fully validating the effectiveness of this approach requires significant computational resources, such as re-running RLHF and conducting evaluations multiple times on pruned datasets. Due to these constraints, we have decided to leave the comprehensive empirical validation of this mitigation strategy to future work. We believe this is a promising direction that could systematically address misalignment issues and complement our current findings.
>
> $\textbf{Q4.2 Dataset limitations}$
>
> $\textbf{A4.2}$ This is answered in global response A0.2. Please let us know if you have further questions.
>
> $\textbf{Q4.3 Model scale constraints}$
>
> $\textbf{A4.3}$ This is answered in global response A0.1. Please let us know if you have further questions.
>
> $\textbf{Q4.4 Additional fine-tuning combined with RLHF to mitigate observed trustworthiness degradation}$
>
> $\textbf{A4.4}$
> Yes, this is certainly possible. As proposed by Haider et al. (2024), such post-training requires a “break-fix” cycle, involving multiple rounds of dataset curation, safety post-training, benchmarking, red teaming, and vulnerability identification to address various harm areas systematically. While algorithm-based mitigation strategies are viable, our intuition is that they need to be combined with (or relied on) high-quality and  trustworthiness-aware preference datasets. These datasets provide supervision specifically aligned with trustworthiness goals, serving as a critical foundation for addressing observed degradations after RLHF.
>
> $\textbf{Q4.5 Possibility of using smaller, targeted RLHF datasets or combinations of general-purpose and aspect-specific datasets}$
>
> $\textbf{A4.5}$ This is also answered in global response A0.2.
>
> $\textbf{Q4.6 Influence of dataset size (used for RLHF) on model trustworthiness metrics}$
>
> $\textbf{A4.6}$
>
> While this is not the primary focus of our work, we are happy to share some preliminary thoughts. From a dataset perspective, two key factors influence a model's performance on trustworthiness metrics: (1) the diversity of the fine-tuning dataset, which ensures the model has sufficient generalizability, and (2) the alignment between the dataset and the trustworthiness aspects being evaluated.
> In terms of dataset size, it typically needs to be sufficiently large to achieve adequate diversity. For widely used general-purpose preference datasets, such as Anthropic-HH, diversity is less of a concern due to their broad coverage. However, our findings highlight a misalignment between Anthropic-HH and several trustworthiness metrics, leading to decreased performance on those tasks.
>
> For your question, the impact of dataset size on trustworthiness depends heavily on the level of alignment between the dataset and the trustworthiness aspects. If alignment is strong and computational resources are not a constraint, larger datasets generally lead to better performance by providing more comprehensive training signals. Conversely, if the dataset is misaligned with the trustworthiness dimensions, increasing its size may not improve performance and could even exacerbate issues.
>
> We did not conduct experiments varying dataset sizes, as this would be computationally intensive and not directly related to our main research question. However, future work could explore this relationship more systematically to better understand the trade-offs between dataset size, alignment, and trustworthiness outcomes.

---

> ### Comment · Reviewer_p6uw · 2024-11-25
>
> Thanks for the clarification. I will keep my positive score.

---

### Official Review · Reviewer_zV3P · 2024-11-02

**Soundness:** 3
**Presentation:** 3
**Contribution:** 3
**Rating:** 6
**Confidence:** 3

**Summary:**

The authors systematically assess how two RLHF variants, PPO and DPO, influence these trustworthiness dimensions using general-purpose preference data. What’s more, this paper introduces a data attribution method to analyze the influence of individual fine-tuning data points on trustworthiness, suggesting the need for more nuanced RLHF approaches.

**Strengths:**

1. This study offers a systematic assessment of RLHF on multiple trustworthiness aspects;
2. The introduction of an influence-function-based data attribution method tailored to RLHF offers insights into individual data points’ impacts, aiding future data selection for alignment;
3. The work highlights the limitations of using general-purpose datasets for trustworthiness, emphasizing the importance of targeted data in achieving specific alignment goals.

**Weaknesses:**

1. The experiments are limited to models up to 7B parameters, which may limit the generalizability of findings to larger;
2. The study relies solely on the Anthropic HH dataset, and results might vary with different datasets;

**Questions:**

1. How might these findings change with LLMs that exceed 7B parameters?
2. Would a more specialized preference dataset yield different outcomes for bias or other dimensions compared to the general-purpose Anthropic HH dataset?
3. What guidance do the paper’s evaluation results offer for balancing helpfulness and trustworthiness in the future?

---

> ### Author Response · Authors · 2024-11-19
> **Author Rebuttal for Reviewer zV3P**
>
> We genuinely appreciate your feedback on our paper, and here we would like to address your questions and concerns raised in the review. We also include our responses for common questions shared by multiple reviewers in a separate section above.
>
> $\textbf{Q3.1 Generalizability of the findings to larger models}$
>
> $\textbf{A3.1}$ This is answered in global response A0.1. Please let us know if you have further questions.
>
> $\textbf{Q3.2 More specialized preference datasets}$
>
> $\textbf{A3.2}$ This is answered in global response A0.2. Please let us know if you have further questions.
>
> $\textbf{Q3.3 Guidance for balancing helpfulness and trustworthiness}$
>
> $\textbf{A3.3}$
> Relevant discussions are briefly included in Sections 5 and 6, but we are happy to expand on them in the updated version. Broadly speaking, addressing the limitations of RLHF requires a more nuanced approach than simply applying it to crowdsourced, general-purpose datasets. For example, using datasets tailored to individual trustworthiness aspects (as some reviewers have suggested) might be able to improve alignment with specific goals. Additionally, performing dataset pruning (e.g. using our proposed data attribution method) might help identify and remove samples likely to negatively impact trustworthiness dimensions. These strategies offer promising avenues to enhance alignment while mitigating unintended consequences.

---

> ### Comment · Reviewer_zV3P · 2024-11-25
> **Official Comment by Reviewer zV3P**
>
> Thanks for the explanation, I will keep my positive score.

---

### Official Review · Reviewer_Hf3B · 2024-11-03

**Soundness:** 3
**Presentation:** 4
**Contribution:** 3
**Rating:** 8
**Confidence:** 4

**Summary:**

**Summary**:
This paper investigates the overall impact of RLHF training on the moral values exhibited by LLMs. The analysis is carried out by fine-tuning base models (up to 8B in size) in general-purpose preference alignment data and then contrasting the performance before and after RLHF across 5 trustworthiness tenets: toxicity, social bias, ethics, truthfulness, and privacy. The obtained results suggest that although RLHF improves ethics measurements by 16.5-18.5% points on average, it significantly worsens performance in other verticals: social bias, truthfulness and privacy leakage worsen on average by 150%, 25% and 12%, respectively.

**Main contributions**:
The main contribution is the idea that RLHF models trained on general-purpose preference data may not align with other trustworthiness values that we care about. The application of data influence functions is also novel and interesting.

**Methodology**:
- The authors start with the Anthropic Harmlessness and Helpfulness (HH-RLHF) preference alignment dataset, from which they fine-tune the base model (SFT), the reward model (RM), as well as two RLHF-based models (one trained with DPO loss and the other with PPO loss).
- Adopting the evaluation testbed proposed in [Wang et al 2024](https://arxiv.org/pdf/2306.11698), the authors report metrics for the four different variants of the models (base, SFT, DPO, PPO) and across model sizes (1.4B up to 8B) across 5 trustworthy tenets contrasting the impact of fine-tuned models with base models.
- Finally, the authors use data influence functions to determine the impact of individual data points in the HH-RLHF dataset for each of the trustworthiness dimensions.

**Interpretation of Results**:
- The results reported across 5 different trustworthiness benchmarks show that fine-tuning LMs in general-purpose preference data (RLHF-HH) leads to worse model behavior in 4  benchmarks when compared to not fine-tuning in RLHF-HH. Across four models, RLHF leads to average improvements of up to 18.5% in recognizing morally wrong scenarios (Figure 3 left). For all other benchmarks, the metric values decrease on average across models by at least 12% points (in Figure 4 left) up to 30% (Figure 3 right).
- In lines 228-229, the authors claim that “toxicity first increases notably after SFT“ which suggests that there is a large improvement in toxicity. However, there is no statistical test supporting this claim and the error bars are overlapping for two larger models (Figure 2 left), suggesting non-significant differences resulting from the SFT procedure.
- In Section 4.2 the authors claim that “both PPO and DPO significantly increase the stereotypical bias scores [...] and the SFT step is most responsible for this increase.”. However, the carried evaluation consists (to the best of my knowledge) of prompting models whether they agree with a stereotypical sentence concerning a marginalized group but not whether the model agree with the same stereotypical sentence concerning a different group. In other words, **the reported results do not disentangle the models’ propensity to agree with claims (as suggested in lines 260-263 and by prior work (Sharma et al 2023)) from the models’ stereotypical bias**.
- In Section 5, the authors mention that “Table 5 clearly demonstrates that the language model becomes increasingly deterministic through RLHF, which may provide some insights on why PPO and DPO fail to decrease toxicity [...]”. However, the maximum log likelihood difference between base model and SFT is only 0.78, raising questions about whether this difference is due to noise or if it’s a systematic behavior. Moreover, the generations for the original toxicity experiment were obtained using temperature=0.5, which modifies the actual distribution of the SFT model, potentially leading to narrower distributions. It is important that the authors clarify which temperature was used to carry the experiments in Appendix F.


### Writing
The paper could improve its clarity by adding relevant citations, better positioning their work with respect to previous work, and being clear about the hyperparameters used during the evaluations. Consider the following suggestions:
- Relevant citations: overall the authors would benefit from adding more relevant citations to back their arguments and strengthen the motivation.
- In the Related Work, subsection “Reinforcement Learning Human Feedback section”, the authors could add citations to the RLHF-HH dataset, but also to other preference datasets (e.g., Cui et al, 2023, Ethayarajh et al 2022). It could also be useful to motivate why the authors settled on the RLHF-HH dataset as opposed to other datasets.
- Lines 260-263 could cite work reporting sycophancy and language model (Sharma et al 2023).

**Clarity**:
- Line 132: “D_demo” → “D” (to be consistent with Equation 1)?
- Equation 1: consider specifying what y_w right after Equation 1 or at the beginning of the section.
- The authors should try to clarify the differences between their experimental design and that of Havrilla et al (2023) and Wang et al (2024). For instance, the analyses carried in Section are very similar to the ones reported in Wang et al (2024), in particular, the evaluation in Section 4.2 appears to be the same but no citation is provided. It could also be nice to motivate the selection of GPT-J-6B based on Havrilla et al (2023) in case the authors didn’t perform any additional experiment using reward models.
 - The authors use different hyperparameters depending on the evaluated criteria. While these parameters are specified in the Appendix C, the only hyperparameters mentioned in the main paper are those for the toxicity evaluation (without referring to the Appendix C or differences in the hyperparameters). The authors should consider being more explicit within each section, motivating the reasons behind selecting different hyperparameters. This is relevant as different hyperparameters have different implications (e.g., lower temperature values have been shown to be associated with higher toxicity)
- Could be useful to add examples for each of the evaluation datasets. Specifically, for the Steoreotypical Bias (in Section 4.2), it would be useful to get a sense of what are the demographic groups, their cardinality, and also the types of stereotypes explored.
- Could be useful if the authors provided a notion of what a meaningful influence score is.
- Figure 5 caption could be more informative, providing more insights on the performed analyses.

Typos:
- Line 398: “8 with 9” → “Equation 8 with Equation 9”
- Line 407: “in 9” → “in Equation 9”

### References

- Touvron et al (2023): Llama 2: Open Foundation and Fine-Tuned Chat Models (https://arxiv.org/abs/2307.09288)
- OpenAI Team (2023): GPT-4 Tech Report (https://arxiv.org/pdf/2303.08774)
- Cui et al (2023) UltraFeedback: Boosting Language Models with Scaled AI Feedback (https://arxiv.org/abs/2310.01377)
- Ganguli et al (2022): Red Teaming Language Models to Reduce Harms: Methods, Scaling Behaviors, and Lessons Learned (https://arxiv.org/abs/2209.07858)
- Ethayarajh et al (2022): Understanding Dataset Difficulty with V-Usable Information (ICML 2022). (https://proceedings.mlr.press/v162/ethayarajh22a.html)
- Askell et al (2021): A General Language Assistant as a Laboratory for Alignment (https://arxiv.org/abs/2112.00861)
- Glaese et al (2022): Improving alignment of dialogue agents via targeted human judgements
- Tirumala et al (2023): D4: Improving LLM Pretraining via Document De-Duplication and Diversification (https://arxiv.org/abs/2308.12284)
- Sharma et al (2023): Towards Understanding Sycophancy in Language Models (https://arxiv.org/abs/2310.13548)

**Strengths:**

- Important research question regarding whose preferences concerning what behaviors models are implicitly learned through RLHF.
- The finding that privacy leakage increases with RLHF and is due to the DPO/PPO algorithms (as opposed to SFT) is interesting.
- The use of data influence functions to traceback the impact of RLHF training data to different aspects of trustworthiness is novel and interesting and could prove beneficial to inform the creation of higher quality and better-aligned preference datasets.

**Weaknesses:**

- Unclear contribution with respect to previous work: Wang et al (2024), GPT-4, and Llama 2 paper tech reports also carry analyses assessing the impact of RLHF in bias, toxicity, and truthfulness. The authors should make their contribution clearer and compare with previous work.
- The evaluation is carried out in Pythia and Llama models as opposed to more recent and prominent models like OLMo and Llama 3 models. Given the considerable differences between the models (e.g., data deduplication and diversity (Tirumala et al 2023)) it is unclear how the results will generalize to these newer and carefully trained base models.
- The reported analysis using influence functions provides no insights into the observed patterns in RLHF or the optimal composition of the RLHF training data. It could be beneficial to add a section with further analysis and/or a discussion section about how the authors see this analysis being useful in the long-term to fix the mis-alignment problem.

**Questions:**

1. One of the main claims in the paper is “[RLHF]’s assumed effect on model trustworthiness hasn’t been rigorously evaluated” (for instance lines 15-16 and 42-43). However, the evaluation of  RLHF impact in ethical aspects (e.g., truthfulness, bias, toxicity, ethics) has been reported previously (Askell et al 2021, Glaese et al 2022, OpenAI Team 2024, Touvron et al 2023). Can you provide a more detailed comparison of your methodology and findings with the previous works, highlighting specific differences or extensions to their approach?
2. In lines 202-203, the authors mention “since our language models are large enough, we use zero-shot generated outputs for all benchmarks”. However, a limitation of base models is that they struggle to adapt to the output format. Specifically they may struggle to adhere to the multiple-choice question format of the TruthfulQA (Section 4.4) or to produce yes/no/unknown for the stereotypical bias experiment (Section 4.2). How do you ensure that the output of the language model is consistent with the multiple choice question answering format? Do you perform any test to assess the capability of LLMs to handle 0-shot QA? You could include a specific analysis of the models' ability to adhere to the required output formats in zero-shot settings, perhaps by reporting the percentage of responses that correctly follow the format for each task.
3. The toxicity evaluation reported in line 222 differs from the original evaluation procedure (5 independent generations and temperature=0.5 vs 25 generations and temperature=1). The lower number of generations may substantially affect the results and the lower temperature modifies the model’s distribution, potentially, skewing results. Could you please motivate the selection of these parameters for your analysis? Moreover, you could provide a sensitivity analysis showing how your results change with different numbers of generations and temperature settings. This would help clarify the robustness of your findings.
4. Some experiments like the stereotypical bias (Section 4.2) are carried out by eliciting whether the model agrees or not with a specific stereotypical topic. How can we guarantee that the observed bias amplification is not an artifact of the evaluation (e.g., models being more likely to output yes vs no, or their sycophantic tendency (Sharma et al 2023))?
5. In Section 4.3, the authors mention that they aim to measure a model's ability to detect morally wrong actions by prompting models whether each example is against human morality. This presupposes that the correct answer (true positive) is to answer yes and that a wrong answer is to answer “no” (false negative). However, the authors call their metric FPR, which does not match with this intuition. Can you provide a clear definition of a true positive, true negative, false positive, and false negative are in this context and explain why FPR is the most appropriate metric for your analysis?
6. What exactly is the prompt being used to prompt base models? If you’re using different prompts, are models sensitive to the prompts, i.e., do you observe considerably different behaviors depending on the prompt used?
7. The privacy evaluation is unintuitive (to me). I am not surprised that fine-tuning LLMs with general purpose preference data would lead to improvements in such a task. In particular, (1) there is no explicit fine-tuning stage that could boost models’ ability to understand the instructions, (2) most conversation scenarios that I can think of the information shared by the user is already known by it. Would you mind commenting on the motivation and practicality of the proposed evaluation?
8. In lines 365-366, the authors mention that “the most intuitive answer is that the HH dataset used for RLHF is out-of-domain for the evaluation tasks” but provide no support for this hypothesis. Have you considered reporting the perplexity of the evaluation datasets  and comparing it to the perplexity achieved in the training dataset or the Ethics benchmark?

---

> ### Author Response · Authors · 2024-11-19
> **Author Rebuttal for Reviewer Hf3B (Part 1)**
>
> We appreciate your detailed review and valuable feedback for our paper, and here we would like to address the main questions and concerns raised in the review. We also include our responses for common questions shared by multiple reviewers in a separate section above. Additionally, we will incorporate presentation-related suggestions into the updated paper as much as possible.
>
> $\textbf{Q2.1 Generalizability to newer models like OLMo and Llama 3 models}$
>
> $\textbf{A2.1}$ This is answered in global comment A0.1. Please let us know if you have further questions or comments.
>
> $\textbf{Q2.2 Insights into the data attribution results and mitigation strategies for the misalignment issue}$
>
> $\textbf{A2.2}$
>
> As we mentioned in our response A1.2, it’s hard to make generalized claims about the observed patterns of contribution scores shown in Figure 5 and Figure 11 without considering the specific model and the current RLHF step, as the trends vary across different configurations.
>
> As partially hinted in our intro and conclusion sections, a key application of our data attribution approach is its potential to identify influential data points that negatively impact specific trustworthiness aspects. These data points can then be removed from the RLHF dataset (i.e. dataset pruning) to address the misalignment issue. Similar discussions are also included in A4.1, A4.4, and A4.6. However, rigorously testing the effectiveness of this approach would require re-running RLHF and trustworthiness evaluations multiple times, which is computationally prohibitive and also not directly related to the focus of this work. For this reason, we have decided to leave such empirical validations for future research.
>
> The main contribution of this work is to shed light on the misalignment problem in RLHF and provide data-level insights through our attribution analysis. We recognize that further exploration is needed to fully leverage this approach, and we will add more discussion on its potential applications and long-term utility in addressing misalignment in the updated paper. We greatly appreciate your valuable feedback.

---

> > ### Comment · Reviewer_Hf3B · 2024-11-21
> >
> > Thank you for putting the effort in addressing my comments. As you point out, the observed patterns depend on the different models and depending on whether you're building an RLHF dataset to be used by different models or for a single model, you may use the insights from your data attribution section differently. For that reason, I think it is important to discuss in the paper how  you envision this type of insights being used to build better RLHF datasets.

---

> ### Author Response · Authors · 2024-11-19
> **Author Rebuttal for Reviewer Hf3B (Part 2)**
>
> $\textbf{Q2.3 Contributions compared with prior works}$
>
> $\textbf{A2.3}$
>
> We consider our evaluation to be “the first systematic evaluation of RLHF’s impact on key trustworthiness aspects.” By this, we mean:
>
> 1. It uses the most widely adopted general-purpose RLHF dataset (Anthropic-HH), popular open-source models, and standard RLHF procedures.
>
> 2. It provides clear, stage-wise comparisons of model trustworthiness across different RLHF steps (SFT, PPO, and DPO).
>
> 3. It evaluates five trustworthiness aspects using standard, widely accepted benchmarks, enabling direct comparisons.
>
> Here we’d also like to highlight key differences between our evaluation and each previous work you referenced above:
>
> 1. Askell et al. 2021: This work focuses on SFT and separate ranked preference modeling tasks, rather than standard RLHF workflows used nowadays. Moreover, it does not include comprehensive trustworthiness evaluations—its overlap with our work is limited to some ethics benchmarks.
>
> 2. Glaese et al. 2022: The goal of this work is to build a highly aligned dialogue agent by collecting preference data with fine-grained and targeted rules (23 rules in total), which is quite different from our general-purpose preference dataset. While some rules partially overlap with our trustworthiness aspects, their approach fundamentally differs by augmenting standard RLHF with multiple reward models designed for these specific rules. Our goal is completely different: to identify misalignments between a general-purpose dataset and trustworthiness aspects without such targeted augmentations for RLHF.
>
> 3. Wang et al. 2024: This study evaluates trustworthiness on closed-source GPT-3.5 and GPT-4 models without explicit RLHF experiments, making direct before-and-after RLHF comparisons impossible.
>
> 4. GPT-4 Tech Report (2024): The RLHF dataset is proprietary, and while the report compares base and fine-tuned models, the focus is not on trustworthiness aspects. Discussions around safety primarily involve comparisons against earlier model versions, not pre-RLHF baselines.
>
> 5. LLama 2 (Touvron et al 2023): Their internal RLHF dataset is proprietary and incorporates fine-grained supervision (e.g., safety labels), which differs from the general-purpose Anthropic-HH dataset used in our study. Additionally, trustworthiness evaluations in their work didn’t include critical aspects like machine ethics and privacy leakage.
>
> In addition to much more controlled trustworthiness comparisons before and after RLHF when compared with prior works, our work has another novel contribution: adapting data attribution methods to the RLHF setting. This approach provides unique, data-level insights into the relationship between training data and trustworthiness outcomes. We will further emphasize these distinctions and contributions in the introduction and related work sections to clarify how our methodology and findings extend previous research. Thank you for your valuable feedback.
>
> $\textbf{Q2.4. Are the base models capable enough to follow task-specific instructions during evaluation?}$
>
> $\textbf{A2.4}$
>
> Thank you for pointing this out. Our decision to use 1.4B as the smallest model size stems from our observation that this size is the minimum required for consistent adherence to output formats across the evaluation tasks. Specifically:
>
> Toxicity and privacy tasks: These do not require explicit generation formats, so adherence is not an issue.
>
> Bias and ethics tasks: Models are prompted to respond with “Yes” or “No,” followed by reasoning. All four models reliably follow this format for all evaluated prompts (i.e. 100% success).
>
> Truthfulness evaluation: This is the only task where we observed occasional format inconsistencies. In this task, the model is presented with a multiple-choice question and instructed to repeat the correct answer explicitly. Failures occur when the model does not repeat any of the provided options. We report the percentage of base model generations that correctly adhere to this instruction in the table below.
> These observations indicate that the base models are generally capable of following task-specific instructions even in zero-shot settings, and we will add these details to the appendix.
>
>
> |    | Pythia-1.4B     | Pythia-2.8B     | Pythia-6.9B     | Llama-7B     |
> |----------|----------|----------|----------|----------|
> |Percentage of correct answer format (Truthfulness) |  91.8  | 97.3 | 97.8 | 100 |

---

> > ### Comment · Reviewer_Hf3B · 2024-11-21
> >
> > Again, thank you for the detailed explanation and for running the additional experiments.
> >
> > In its current version, I believe that the paper didn't distinguish itself enough from other existing studies. I think positioning the paper by clearly stating its contributions wrt to prior research makes it a better motivated and well-supported paper.

---

> ### Author Response · Authors · 2024-11-19
> **Author Rebuttal for Reviewer Hf3B (Part 3)**
>
> $\textbf{Q2.5 Different generation configs compared with prior work in toxicity evaluation}$
>
> $\textbf{A2.5}$
>
> The specific configuration of 25 generations and temperature = 1, as used by Wang et al. (2024), is tailored for evaluating very large models like GPT-3.5 and GPT-4. Since our models are significantly smaller, we do not believe their output distributions are directly comparable. Moreover, it is unclear why maintaining the same distribution as a prior work with different models would be necessary (or could you clarify why this is important?). As long as we consistently apply the same generation configuration across all toxicity evaluations, our experiments remain in a highly controlled setting.
>
> The prior work (Wang et al., 2024) also relied on a single model configuration for toxicity evaluation. We believe the robustness of our findings is further supported by the fact that we took the average across 1,200 prompts. Our choice of num_generations=5 is primarily due to computational constraints and the rate limits of Perspective API calls. To comply with these limits, we also opted for a lower temperature of 0.5 to produce a more centralized distribution.
>
> Since our focus is on expected maximum toxicity (i.e. the maximum among the 5 generations) rather than the mean, increasing the number of generations from 5 to 25 would not enhance the robustness of the results. Under this metric, toxicity may slightly increase (with diminishing returns) as the number of generations or the temperature value increases. However, this expected increase does not compromise the robustness of our claims. A highly systematic sensitivity analysis would be time-consuming and not feasible within the discussion period, and in our opinion, it is not strictly necessary.
>
> $\textbf{Q2.6 Sycophantic tendency of language models}$
>
> $\textbf{A2.6}$
>
> This is indeed an interesting question to discuss. We agree that the observed model behaviors (especially in bias and ethics) could be potentially decoupled into true model understanding and it’s inherent tendency to agree with user beliefs, but we’d like to argue that sycophancy in language models is not merely an artifact of evaluation, but a reflection of the model's alignment with user expectations. In human interactions, individuals often conform to social norms and expectations to build rapport and trust. Similarly, LLMs are designed to align with user inputs, which can manifest as agreement with presented statements. Consider a scenario where a person is asked, "Do you think people from [specific group] are less competent?" Even if the individual does not hold this belief, social pressures or a desire to conform might lead them to agree, especially if they perceive that agreement is expected or will be favorably received. This human tendency to conform or agree, even against personal beliefs, parallels the sycophantic behavior observed in LLMs. In other words, this component is part of the model's trustworthiness behavior, and we should not exclude it in our evaluation, as is consistent with prior works like Wang et al. (2024).
>
> As a side note, the decoupling to evaluate true model beliefs could be done by using a balanced set of user prompts (e.g. constructing the statements with opposite meanings, or simply taking the average between “do you agree…” and “do you disagree”), and this would also be an interesting follow-up work for future research that requires rigorous experimental design.
>
> $\textbf{Q2.7 Definition of FPR in machine ethics evaluation}$
>
> $\textbf{A2.7}$
>
> We appreciate the reviewer’s observation and agree that our current framing of the false positive rate (FPR) as the primary metric could be confusing given the context. The evaluation setup we referenced from Wang et al. considers morally neutral or positive actions as the “positive” class and immoral actions as the “negative” class. This led us to adopt FPR for consistency. However, we acknowledge that our goal is to evaluate the model’s ability to correctly identify morally wrong actions, so it’s more appropriate to switch the definitions of positive and negative cases and then call our metric “false negative rate” (FNR). FNR reflects the rate at which morally wrong actions are mistakenly overlooked, which is critical because failing to recognize such actions undermines the model’s trustworthiness in ethical decision-making tasks.

---

> > ### Comment · Reviewer_Hf3B · 2024-11-21
> >
> > > The specific configuration of 25 generations and temperature = 1, as used by Wang et al. (2024), is tailored for evaluating very large models like GPT-3.5 and GPT-4. Since our models are significantly smaller, we do not believe their output distributions are directly comparable.
> >
> > The specific configuration of 25 generations and temperature = 1 was originally used in the [RealToxicityPrompts paper](https://arxiv.org/pdf/2009.11462) (back in 2020) when neither GPT-3.5 nor GPT-4 existed, so I don't think the argument _"Since our models are significantly smaller, we do not believe their output distributions are directly comparable."_ holds.
> >
> > > Moreover, it is unclear why maintaining the same distribution as a prior work with different models would be necessary (or could you clarify why this is important?). As long as we consistently apply the same generation configuration across all toxicity evaluations, our experiments remain in a highly controlled setting.
> >
> > Setting temperature=0.5 skews the distribution, encouraging the sampling of more likely tokens (in doing so, there's an assumption that the models will be used in that same way). As a result, it may not be as representative of the whole spectrum of model behavior, especially if you evaluate such behavior in a small number of samples. I agree with you that 1.2k samples is large enough of a sample. Note however, that it's usually good practice to motivate the choices of these parameters (especially when deviating from the original proposed methodology).
> >
> > > Since our focus is on expected maximum toxicity (i.e. the maximum among the 5 generations) rather than the mean, increasing the number of generations from 5 to 25 would not enhance the robustness of the results. Under this metric, toxicity may slightly increase (with diminishing returns) as the number of generations or the temperature value increases. However, this expected increase does not compromise the robustness of our claims. A highly systematic sensitivity analysis would be time-consuming and not feasible within the discussion period, and in our opinion, it is not strictly necessary.
> >
> > Actually, I agree that such analysis is not a priority or even necessary. However, I still encourage that the authors mention the resource constraints as the reason for selecting different hyperparameters for generation.

---

> ### Author Response · Authors · 2024-11-19
> **Author Rebuttal for Reviewer Hf3B (Part 4)**
>
> $\textbf{Q2.8 Sensitivity to different prompts}$
>
> $\textbf{A2.8}$
> For specific examples of prompts being used in our evaluations, please refer to appendix A. We did evaluate the model sensitivity to different system prompts for toxicity, stereotypical bias, and machine ethics tasks, and the results are reported in appendix D and E. For toxicity, we also evaluated on toxic and non-toxic user prompts, and the details can be found in section 4.1. The general observation is that the trends are consistent across all these different settings. For truthfulness evaluation, we experimented with different prompting strategies and found most success with directly asking the model to repeat its answer among all multiple-choice options. We used the same prompt for all different model configurations.
>
>
> $\textbf{Q2.9 Motivation and practicality of the privacy evaluation}$
>
> $\textbf{A2.9}$
> We included this privacy evaluation in our work because it exemplifies cases where helpfulness and trustworthiness/safety are in direct conflict, based on human intuition. While RLHF fine-tuning seeks to make models more helpful and aligned with user instructions, it can inadvertently increase compliance with harmful requests, such as disclosing sensitive information. Evaluating privacy leakage underscores this trade-off, showing how prioritizing helpfulness without explicitly addressing safety risks can undermine trustworthiness. These scenarios are highly relevant in real-world applications, where users often share private data with the expectation that the system will safeguard it.
>
>
> $\textbf{Q2.10 Verification for the claim that “the HH dataset used for RLHF is out-of-domain for the evaluation tasks”}$
>
> $\textbf{A2.10}$
>
> Thank you for your suggestion. In the table below, we report the changes in average perplexity scores across all evaluated prompts for five trustworthiness tasks after the initial SFT with the Anthropic-HH preference data, using Pythia-6.9B. The results show that perplexity scores for all five evaluation tasks slightly increase after SFT. This indicates that there are indeed out-of-domain (OOD) issues between the fine-tuning data and the evaluation datasets, even for the ethics benchmark, where we observe improvements in model trustworthiness.
>
> |          | Anthropic-HH (static) | Toxicity | Bias | Ethics | Truthfulness | Privacy |
> |----------|----------|----------|----------|----------|----------|----------|
> | Before SFT  | 8.98 | 85.1 | 48.7 | 68.6 | 60.4 | 40.9 |
> | After SFT  | 7.69  | 90.7 | 50.1 | 72.5 | 63.0 | 42.2 |
>
> That said, we would like to clarify that while perplexity is a useful measure for identifying surface-level OOD issues (e.g., unfamiliar vocabulary or syntax), it may not adequately capture deeper alignment or misalignment between datasets from a trustworthiness perspective. By describing the HH dataset as "OOD for evaluation tasks," we mean that it may include samples that are not beneficial—or may even be detrimental—to specific trustworthiness aspects. This higher-level misalignment is likely beyond what perplexity scores alone can reveal.
>
> To better illustrate this OOD issue, particularly from a trustworthiness perspective, we have provided concrete examples of fine-tuning samples that negatively impact evaluation tasks. These examples are included in Figure 6.
>
> $\textbf{Q2.11 Statistical significance of differences in log likelihood values reported in Table 5}$
>
> $\textbf{A2.11}$
> Thank you for pointing this out. These log-likelihood values are essentially negative perplexity scores, but they are computed on the model's self generations rather than a given corpus of texts. While the changes in these values might appear small, they are statistically significant, especially given that they are averaged over thousands of prompts and five generations per prompt. We will include the confidence intervals based on these five generations in the updated paper to better show the statistical robustness of this particular analysis.
>
> $\textbf{Q2.12 Statistical significance supporting the claim “toxicity first increases notably after SFT"}$
>
> $\textbf{A2.12}$
> Thanks for pointing this out and you are right that we should not use the phrase “increases notably” here based on the overlapping error bars. We will make this revision in the updated paper.

---

> > ### Comment · Reviewer_Hf3B · 2024-11-21
> >
> > Thank you.
> >
> > I'd encourage the authors to clarify what they mean by "OOD for evaluation tasks" in the camera-ready version of the paper.

---

> ### Comment · Reviewer_Hf3B · 2024-11-21
>
> As for the general comment on why you don't include newer models, I don't fully agree with the claim "We chose Pythia and older Llama models because they are most widely used within the research community, supporting reproducibility and cross-comparisons." In particular, OLMo won the best theme paper award in ACL which emphasizes the importance of OLMo for the research community. Perhaps the authors can convince me by providing examples of uses of Llama or Pythia models in the community that are still widely used and relevant in the research community.
>
> However I do agree that the core principles of RLHF and the findings of this paper remain relevant across architectures and that performing such studies across multiple models would be infeasible.
>
> Based on the discussed changes to the camera ready version, I'm raising my score as I believe this work is important for the community and the authors addressed the weaknesses and questions.

---

> > ### Author Response · Authors · 2024-11-25
> >
> > Thank you for your thoughtful replies and for acknowledging our rebuttals.
> >
> > Yes, we agree that it would be helpful to apply our evaluations to newer models like OLMo and Llama 3, which were not as popular when we started the experiments earlier this year. We will consider adding additional results for these models in our camera-ready version if this paper is accepted.
> >
> > Again, we greatly appreciate your detailed review and constructive advice for this paper.

---

### Official Review · Reviewer_AQ8v · 2024-11-04

**Soundness:** 4
**Presentation:** 4
**Contribution:** 3
**Rating:** 8
**Confidence:** 3

**Summary:**

The paper evaluates how Reinforcement Learning from Human Feedback (RLHF) impacts Large Language Model (LLM) trustworthiness. It assesses five dimensions: toxicity, bias, machine ethics, truthfulness, and privacy. Results show mixed outcomes: while RLHF improves machine ethics, it often increases bias and privacy leakage and reduces truthfulness, with minimal impact on toxicity. The paper also introduces a novel data attribution method in the RLHF setting to analyze RLHF's effect on these dimensions, highlighting the need for more targeted alignment strategies.

**Strengths:**

- The paper provides a detailed analysis of RLHF methods across multiple trustworthiness dimensions, offering nuanced insights into alignment challenges.
- By comparing SFT, PPO, and DPO, the paper highlights strengths and weaknesses of each method on its effect on trustworthiness, helping guide future model alignment choices.
- The paper evaluates trustworthiness through several dimensions—such as bias, truthfulness, toxicity, ethics, and privacy—rather than limiting its analysis to one aspect. This multi-faceted approach strengthens the research by showing how alignment impacts different areas, which are crucial for real-world applications where trustworthiness encompasses various dimensions.
- The proposed data attribution method is useful for observing which data points most significantly affect certain trustworthiness dimensions, such as reducing toxicity. By attributing changes in model behavior to specific data inputs, this method provides insights into how particular data segments influence alignment outcomes.
- The paper is well-written and easy to follow.

**Weaknesses:**

- In this paper, toxicity is evaluated using the Perspective API. However, several existing papers (see https://aclanthology.org/2023.emnlp-main.472.pdf and https://arxiv.org/pdf/2312.12651) have raised concerns that such black-box APIs may introduce inaccuracies or biases in toxicity assessments. This may affect the reproducibility of this research on this dimension.
- What are the potential trade-offs between the 5 dimensions in trustworthiness? In real-world applications, improving one dimension (e.g., reducing bias) might compromise another (e.g., truthfulness). Would appreciate some insights here.

**Questions:**

See above weaknesses.

---

> ### Author Response · Authors · 2024-11-19
> **Author Rebuttal for Reviewer AQ8v**
>
> We appreciate your detailed review and recognition of our paper's contributions.  Our responses to your questions and concerns are provided below. We also include our responses for common questions shared by multiple reviewers in a separate section above.
>
> $\textbf{Q1.1 Noises in the black-box API for toxicity evaluation}$
>
> $\textbf{A1.1}$ We acknowledge that inherent noise exists in toxicity evaluations using black-box APIs like Perspective API, and we appreciate you pointing this out. The two papers cited in the review highlight two specific sources of noise: (1) discrepancies in toxicity scores for the same model generations over extended periods (e.g., 2020 vs. 2023), likely due to API updates, and (2) differences across languages (e.g., German translations receiving higher toxicity scores). However, these issues are unlikely to affect our experiment results because (1) all our toxicity evaluations using Perspective API were conducted within approximately two weeks, minimizing the impact of potential API updates, and (2) the RealToxicityPrompts dataset consists entirely of English prompts, eliminating cross-language inconsistencies.
>
> To the best of our knowledge, Perspective API remains the most widely used toxicity evaluation tool in the research community. While human evaluators might seem like an alternative, they are more prone to errors for such rating tasks, and their judgments are also subject to variability over time and across languages. Given these factors, we believe Perspective API is probably the most reliable and practical option for our study.
>
> $\textbf{Q1.2  Potential trade-offs between the five trustworthiness aspects}$
>
> $\textbf{A1.2}$ We believe it’s challenging to make high-level claims about trade-offs between the five dimensions of trustworthiness without considering specific scenarios. In this work, the impact on any trustworthiness aspect depends on the fine-tuning data, the target model, and the specific RLHF step (SFT, PPO, or DPO). Figures 5 and 11 show the aggregated effects across the entire fine-tuning dataset, highlighting these dependencies.
> For any individual trustworthiness aspect, typically only a small fraction of the data points are relevant, so it’s more meaningful to focus on specific data samples or subsets based on estimated contribution scores. For example, we can make microscopic observations like “this subset of fine-tuning data positively impacts bias but negatively impacts truthfulness.” However, this does not imply that bias and truthfulness are inherently in conflict.
> Ultimately, such trade-offs are highly contextual, and generalizing without grounding the discussion in specific data or scenarios risks oversimplification. We believe a data-driven, granular approach is the most effective way to analyze these dynamics.

---

> > ### Comment · Reviewer_AQ8v · 2024-11-26
> > **Response to Authors**
> >
> > Thanks for the clarifications. I will keep my positive score.

---

### Author Response · Authors · 2024-11-19
**Addressing Common Questions Shared by Reviewers**

We genuinely appreciate the reviewers for their valuable and insightful comments. In this section, we would like to address several common concerns raised by the reviewers:

$\textbf{Q0.1. Generalizability to large/newer models}$

$\textbf{A0.1}$

Regarding model size:

Our experiments focus on models up to ~7B parameters mainly due to computational constraints, as we used full-parameter RLHF. Despite this, we expect similar trends for larger models. The evaluation tasks we used do not demand very complex reasoning, and our models (1B–7B) already demonstrate sufficient ability to follow user instructions and generate consistent outputs. Moreover, prior research (e.g. Wang et al. 2024) suggests that larger models are not inherently more trustworthy on the four aspects where we observed negative RLHF effects. Given the trustworthiness trends identified in our experiments, it is reasonable to expect these trends to generalize to larger models.

Regarding newer models:

We chose Pythia and older Llama models because they are most widely used within the research community, supporting reproducibility and cross-comparisons. While newer models may incorporate advanced training techniques, the core principles of RLHF, how it operates on the pairwise datasets, and its impact on trustworthiness metrics remain relevant across architectures. We’d be happy if this work could serve as a foundation for evaluating RLHF effects on more recent models in future research.


$\textbf{Q0.2. Why Anthropic-HH and generalizability to different (e.g. more specific) datasets}$

$\textbf{A0.2}$

First of all, we’d like to clarify that our major claim in this work is “using general-purpose preference datasets for RLHF is problematic from a trustworthiness perspective”, instead of “RLHF universally harms trustworthiness with different fine-tuning datasets”. And our finding is important because using RLHF with general-purpose datasets has become a common practice for model alignment nowadays.

Anthropic-HH, the most widely used general-purpose RLHF dataset, aims to directly align models with helpfulness and harmlessness, making it the best choice for our study. It’s possible that specific datasets tailored to individual trustworthiness aspects might improve performance in those areas, but it remains a fact that current RLHF practice heavily relies on general-purpose datasets like Anthropic-HH, mainly because (1) there aren’t many high-quality pairwise preference datasets publicly available for RLHF and (2) it’s relatively rare to only align models on one specific trustworthiness aspect.

That being said, one takeaway from our study could be that we should consider developing and using task-specific datasets (possibly combining them with general-purpose ones, as reviewer p6uw suggested). These follow-up efforts would need more collaboration in the research community, especially around data collection, and they can’t be addressed in this work alone. Thus, we consider these dataset generalizability concerns as exciting future directions rather than challenges to our conclusions.

$\textbf{Q0.3. Comments on presentation and style}$

$\textbf{A0.3}$
Thanks for all the feedback regarding the presentation of this paper. We will incorporate these changes (e.g. more references, clarification of technical details, etc.) as much as we can to the updated version.

---

### Author Response · Authors · 2024-11-25
**Revised Paper Uploaded**

Thanks again for the valuable feedback from all reviewers. This is just a note that we’ve updated our paper based on the suggestions given by the reviewers. The main changes include:

1. More discussions on our contributions compared with prior works and the generalizability of our findings

2. Results on base model capabilities to follow task-specific answer formats (appendix B)

3. Motivations for language model hyperparameter selection (appendix D)

4. Confidence intervals for results in appendix G

5. More interpretations of data attribution results (section 5 and appendix I)

6. Fixing other minor issues such as inaccurate result descriptions, inconsistent notations, typos, etc.

Again, we greatly appreciate these constructive comments!

---

### Meta-Review · Area_Chair_qT7N · 2024-12-17

**Metareview:**

This paper evaluates four language models Pythia 1.4B, 2.8B and 6.9B as well as Llama-&B on five axes of “trustworthiness” : Toxicity, Stereotypical bias, Machine Ethics, Truthfulness and Privacy.  The main contribution is that optimizing these LLMs for human preferences
does not simultaneously optimize them for trustworthiness, in fact in the aspects of stereotypical bias, truthfulness and privacy it has the opposite effect.

The authors use the Anthropic HH dataset for their evaluation. And show results for: the autoregressive language loss for SFT, the preference loss of the reward model for PPO and the loss of the language model for DPO when compared to the “no RLHF” baseline.

This paper was considered a good paper by all reviewers and in the rebuttal phase some authors increased their score.  It represents an important contribution, and I recommend that it should be accepted.

**Additional Comments On Reviewer Discussion:**

The reviewer discussion was active and unanimously positive (6+).  The Authors responded to authors concerns and in the discussion period two reviewers increased their score.

The concerns included whether or not the results would generalize to newer larger models and whether or not the Anthropic HH dataset was the optimal dataset choice.  Other concerns includes those around "noise" in the toxicity dataset: ambiguity and some statements are mire toxic than others and also whether or not there exist intrinsic tensions between the five aspects of trustworthiness - for example bias and truthfulness.

Overall, while the concerns point out intrinsic limitations that may exist in this area, the reviewers all agreed that this was a worthy contribution and that simply more could be done.

---

### Decision · Program_Chairs · 2025-01-22

Accept (Oral)